# Amphiphilic Oligonucleotide Derivatives—Promising Tools for Therapeutics

**DOI:** 10.3390/pharmaceutics16111447

**Published:** 2024-11-12

**Authors:** Irina A. Bauer, Elena V. Dmitrienko

**Affiliations:** Institute of Chemical Biology and Fundamental Medicine SB RAS, 630090 Novosibirsk, Russia; bauer@niboch.nsc.ru

**Keywords:** modified nucleic acids, amphiphilic oligonucleotides, nucleic acid self-assembly, serum albumin, protein–oligonucleotide complexes

## Abstract

Recent advances in genetics and nucleic acid chemistry have created fundamentally new tools, both for practical applications in therapy and diagnostics and for fundamental genome editing tasks. Nucleic acid-based therapeutic agents offer a distinct advantage of selectively targeting the underlying cause of the disease. Nevertheless, despite the success achieved thus far, there remain unresolved issues regarding the improvement of the pharmacokinetic properties of therapeutic nucleic acids while preserving their biological activity. In order to address these challenges, there is a growing focus on the study of safe and effective delivery methods utilising modified nucleic acid analogues and their lipid bioconjugates. The present review article provides an overview of the current state of the art in the use of chemically modified nucleic acid derivatives for therapeutic applications, with a particular focus on oligonucleotides conjugated to lipid moieties. A systematic analysis has been conducted to investigate the ability of amphiphilic oligonucleotides to self-assemble into micelle-like structures, as well as the influence of non-covalent interactions of such derivatives with serum albumin on their biodistribution and therapeutic effects.

## 1. Introduction

Therapeutic oligonucleotides (tONs) are nucleotide sequences of up to 80 nucleotides in length that possess the ability to hybridise with target DNA or RNA and bind to specific proteins [1,2,3,4]. ON-based drugs are distinguished by their high degree of selectivity in binding to target molecules, which is attributed to the complementarity of the nitrogenous bases of the interacting chains. Consequently, ON-based therapeutic strategies can prevent and treat a range of diseases through the selective inhibition of target gene expression [3,5,6]. Furthermore, the use of ONs as therapeutic agents presents several additional advantages over small-molecule drug compounds. These include a prolonged duration of action and a versatile production technology platform [1,7]. At present, twenty nucleic acid-based drugs have been granted approval for the treatment of a range of human diseases, with over two hundred undergoing preclinical testing [1,6,8,9,10]. Despite the presence of a number of limitations, including the low plasma stability of unmodified tONs, the difficulty of delivery, the high cost of synthesis, and the risk of immunogenic effects, there is a growing trend towards the shift from the use of small-molecule drugs to ON therapy, as it has excellent potential in the fight against poorly treatable diseases [3]. The recent advancements in the synthesis of modified nucleic acid analogues have significantly enhanced their pharmacokinetic properties of nucleic acids. Their modified derivatives are currently regarded as potential tools for addressing a range of issues in the fields of personalised medicine, theranostics, and genomic editing [7,11,12,13,14,15].

The attachment of hydrophobic residues, including cholesterol, fatty acids, lipids, and alkyl-containing groups, to the ON backbone represents a promising approach to enhancing the pharmacokinetic properties of synthetic ONs. The attachment of hydrophobic groups to such derivatives increases their efficiency in penetrating cell membranes and improves their pharmacokinetic properties [3,16,17]. The enhanced pharmacokinetic characteristics and altered biodistribution are typically attributed to the binding of lipid-containing derivatives to serum proteins [18,19]. It is believed that their binding to serum proteins influences their uptake by cells and largely determines their further biodistribution.

Over the last decade, a considerable number of original research papers have been published on the subject of the delivery of tONs to intracellular targets. These papers cover a range of approaches, including chemical modification, bioconjugation, and the use of nanocarriers [1,2,16,20,21]. A considerable number of reviews have been published on lipid-ON conjugates, with a focus on aspects of their synthesis and biological efficacy [3,17,22,23,24,25,26,27]. It is important to note that the authors of these publications do not address the behaviour of such conjugates in solutions, particularly the features of self-assembly into micelle-like structures. The interaction of lipid bioconjugates with serum proteins, including human serum albumin (HSA), has also been extensively studied. However, the existing literature on this topic is disparate, and there are no publications summarising the information presented in various experimental articles.

This review article presents the current state of research in the field of modified nucleic acid analogues, with a particular focus on their bioconjugates with hydrophobic molecules for therapeutic purposes. The majority of the review is dedicated to the generalisation of the most significant information on the potential applications of amphiphilic ON derivatives in therapy. Furthermore, the characteristics of self-assembly and non-covalent interactions with HSA of such ON conjugates are also examined.

## 2. Therapeutic Oligonucleotide Mechanisms of Action

Therapeutic ON-based drugs exhibit considerable diversity. They vary in terms of their mechanism and duration of action, the type of molecule employed, and the cellular location of the target. To date, several classes of tONs have been identified, including antisense ONs (ASO), small interfering RNAs (siRNAs), microRNAs, and aptamers [1,2,3,7,17,28]. Depending on their mechanism of action, tONs can be classified into two main groups for the purpose of inhibiting mRNA functions: steric blocking or the targeted degradation of the relevant mRNA [5]. In the first approach, tONs bind specifically to the target mRNA molecule, sterically inhibiting its interaction with the ribosome, proteins, nucleic acids, or transcription factors. Consequently, the interaction results in either translation arrest (Figure 1c) or the regulation of mRNA maturation (Figure 1b). An alternative mechanism is based on the targeted degradation of the mRNA molecule, which is mediated by specific proteins with nuclease activity. RNase H (Figure 1a) or AGO-2 (Figure 1d). The tON, upon binding to the mRNA, forms a complex that is recognised by the above-mentioned proteins, resulting in the cleavage of the target [1,3]. To date, ten ASO-based drugs and six siRNA-based drugs have been approved for use in clinical practice [6,29,30,31,32].

Aptamers represent a distinct class of molecules, differing from both ASOs and siRNAs. They are single-stranded ONs with a specific sequence that enables them to bind to target proteins with high affinity and specificity (Figure 1e). They are referred to as ‘chemical antibodies’ due to their antibody-like functions, and some have been developed as therapeutic molecules that inhibit proteins associated with infectious diseases or cancer [4]. Currently, MACUGEN^®^ (pegaptanib) is the only aptamer approved by the United States Food and Drug Administration (FDA) [33]. Nevertheless, several aptamer-based drugs are currently undergoing clinical trials [6,34]. In comparison to their protein counterparts, aptamers present a number of advantages, including low immunogenicity, the relative ease of large-scale synthesis, physical stability, and easy chemical modification [6]. Consequently, tONs represent useful tools for the regulation of gene expression and hold considerable therapeutic potential [35].

## 3. Biological Barriers That Determine the Oligonucleotide Pharmacokinetics

The approval of the first ON drug, VITRAVENE^®^, in 1998 marked a significant breakthrough in the treatment of diseases, particularly in the case of cytomegalovirus retinitis. To date, twenty nucleic acid-based drugs have been approved for use in the treatment of various human diseases, eighteen of which are ONs [1,6,8,9,10]. A substantial number of clinical trials have demonstrated the major biological obstacles that influence the pharmacokinetic characteristics of ON drugs, ultimately affecting their therapeutic efficacy (Figure 2). Firstly, nucleases are widely expressed in plasma [36,37]. These enzymes catalyse the hydrolysis of NA by cleaving the phosphodiester bond, resulting in the complete loss of the therapeutic effect of tON before it reaches the target site due to its degradation [1,3,7]. Secondly, the reticuloendothelial system, also known as the mononuclear phagocyte system, is a group of phagocytic cells that act as a form of cellular defence, expelling foreign particles such as toxins, bacteria, and xenobiotics [1,3,7]. Therapeutic ONs do not exclude this phenomenon. Thirdly, tONs are administered exclusively via injection. Due to the thickness of the endothelial tissue, the majority of tONs are retained in the bloodstream and subsequently excreted by the kidneys [1,7]. Consequently, only an indefinitely small quantity is capable of escaping from the vessel lumen and entering the interstitial fluid [1,3,38,39]. Fourthly, ONs have been observed to activate the extracellular and intracellular innate immune responses [40]. Ultimately, the most significant challenge for tON is its limited capacity to penetrate the cell membrane, which is attributed to the hydrophilic nature of the ON and its negatively charged sugar-phosphate backbone [1]. Nevertheless, even after penetration into the cell, the subsequent fate of tON is threatened by destruction in lysosomes [41,42]. It has been demonstrated that therapeutic oligonucleotides, such as siRNA, accumulate in acidic endolysosomal compartments within hours of delivery to primary hepatocytes in vitro and in vivo. The slow release from these compartments appears to delay the onset of activity, and it is of paramount importance to achieve high metabolic stability in order to maximise activity [43,44]. The delivery of ON-based therapeutic agents to the central nervous system presents an additional challenge, as these agents are generally unable to cross the blood–brain barrier [1].

## 4. Chemical Modifications of Oligonucleotides

As the above-mentioned obstacles reduce the efficacy of tONs, approaches to chemically modify them are being actively developed within the framework of solid-phase chemical synthesis in order to increase their delivery efficiency and thus improve their therapeutic properties [6]. Solid-phase synthesis is currently carried out automatically in ON synthesisers, allowing almost any desired ON to be produced quickly and economically in the required quantities [45,46,47,48].

The ON molecule contains several structural fragments that can be modified (Figure 3), namely, the inter-nucleotide phosphate, the ribose backbone, and the nitrogenous base [1,2]. Other potential conjugation sites are the 3′ and 5′ ends of ONs. A substantial number of studies have demonstrated that the modification of the inter-nucleotide phosphate enhances the nuclease resistance of modified ONs in comparison to their unmodified counterparts [49]. The modification of the sugar backbone and nitrogenous bases has been shown to increase the affinity of the therapeutic oligonucleotide for the target sequence [1,6,7,50]. Finally, the introduction of lipid molecules has been observed to facilitate the efficiency of cellular penetration [3,23,25]. Consequently, by combining different modification schemes, it is possible to improve the pharmacokinetics, pharmacodynamics, and biodistribution of an ON drug and ensure its efficient delivery to tissues and organs without the use of additional delivery vehicles [31,41,51,52,53]. The nuclease stability of modified ONs is a critical determinant of its biological activity. Accordingly, in [49], the authors examined the stability of ONs incorporating the major types of modifications, PS and 2′-OMe (Figure 3a,b). In addition to PS and 2′-OMe, a novel class of inter-nucleotide phosphate modifications (PG), which will be discussed in greater detail in Section 4.1, was also investigated. Following a 21-day incubation period in DMEM containing 50% fetal bovine serum, it was demonstrated that the nuclease resistance of modified ONs increased in the following order: deoxy/PS < 2′-OMe/PS << deoxy/PG ≈ 2′-OMe/PG (Table 1).

### 4.1. Phosphate Backbone Modifications

In biological environments, nucleases rapidly hydrolyse phosphodiester bonds in ONs. Consequently, considerable efforts have been made to enhance their nuclease resistance through the introduction of chemical modifications during automated ON synthesis. The first drug to be approved for the treatment of cytomegalovirus retinitis was a phosphate-modified 21-stranded ON in which the oxygen atoms not involved in the phosphodiester bond were replaced by a sulphur atom (PS). In addition to sulphur, other atoms, including carbon, nitrogen, and boron, can be used (Figure 3a). Despite the conceptual simplicity of these changes, they can have significant consequences for the physicochemical and biological properties of the resulting ONs. As an illustration, the replacement of PO with PS has been demonstrated to considerably enhance the pharmacokinetic characteristics of tONs. This is evidenced by an increase in the half-life to 1–2 h and a notable reduction in renal clearance rate [54]. However, at the same time, the binding affinity of such modified ONs to plasma proteins increases, and excessive PS repetition in one unit of tON affects its ability to form a duplex with the target mRNA (melting temperature decreases by approximately 0.5 °C per modification) and leads to the development of off-target toxicity after prolonged exposure [55]. It is noteworthy that, despite the enhanced stability of PS analogues in plasma, they are also susceptible to slow hydrolysis by pyrophosphatase or phosphodiesterase [56]. Nevertheless, despite the above-mentioned disadvantages, PS derivatives have been widely employed in clinical practice [8,31,51,52,53,57,58,59,60,61].

The development of novel modifications of the inter-nucleotide phosphate is ongoing. The replacement of anionic oxygen with carbon leads to the formation of an alkylphosphanate (PC) bond and the production of charge-neutral ON derivatives, which are characterised by low affinity to RNA and are unstable under the basic conditions used to unblock ONs after synthesis [62]. Another well-established modification is the phosphotriester (POR) [63,64]. However, the phosphotriester group has been observed to exhibit reduced stability under conditions typically employed during synthesis and deblocking. The boranophosphate (PB) modification is of particular interest due to the ability of ONs containing such modifications to activate RNase H. Consequently, gapmer ONs containing a boranophosphate modification in combination with a thiophosphate modification were obtained [65]. The Staudinger reaction, which involves the oxidation of phosphite triester by organic azides, has enabled the synthesis of new classes of phosphate-modified ONs, including phosphoryl guanidines (PG) (Figure 4a) [47,48,66]. The combination of this modification with thiophosphate modifications significantly enhances the therapeutic potential of ONs [67]. As an alternative to thiophosphate derivatives, a class of sulfonyl phopshoramidates, ONs (Figure 4b) with the ability to activate RNase H, is being considered [68]. The therapeutic potential of such ONs is currently being investigated at Ionis Pharmaceuticals [69]. In addition, a class of triazinyl phosphoramidates (Figure 4c) ONs has been proposed that allows for flexible variation of substituents introduced across the inter-nucleotide phosphate [70,71].

A noteworthy aspect of introducing a modification into an inter-nucleotide phosphate is the fact that each modified phosphorus atom possesses a chiral centre, which allows for the formation of two distinct stereoisomers (S or R). Consequently, the 20-mer ASO, in which all anionic oxygen has been substituted with sulfur, is a racemic mixture comprising 2^19^ isomeric forms. The physicochemical properties of each stereocenter differ in terms of their hydrophobic/ionic character, resistance to nucleases, affinity for the target molecule, RNase H activity, and siRNA-mediated silencing. It has been demonstrated that the stereopurity of PS-ONs can be effectively regulated to enhance their biological activity [72,73,74,75,76]. Similarly, the introduction of phosphorylguanidine (PG) modifications, with simultaneous control of stereopurity, into both siRNA and ASO has been demonstrated to enhance both silencing efficiency and nuclease stability [67,76]. Furthermore, the activity of siRNA containing PG modifications was found to be dependent on the 2′-modification of the sugar. The introduction of a 2′-F modification to the nucleotide at the 3′-end of the PG bond resulted in greater siRNA inhibitory activity than the 2′-OMe modification [76].

### 4.2. Sugar Backbone Modifications

The ribose residue, particularly its 2′-hydroxyl group, constitutes a crucial component for the recognition of nucleic acids by nucleases and the hydrolysis catalysis of the target mRNA molecule. The combination of DNA (2′-deoxy) and RNA bases is of critical importance for the activity of ASO gapmers involved in RNase H recruitment. This combination is used at the 3′-ends of some siRNAs to impart resistance to nucleases [50]. Similarly, 2′-O-methyl (2′-OMe), 2′-O-methoxyethyl (2′-MOE), and 2′-fluoro (2′-F) (Figure 3b) are among the most commonly used 2′-substituents [1,2,50]. These modifications enhance the resistance of ONs to nucleases by replacing the nucleophilic 2′-hydroxyl group of unmodified RNA. This results in enhanced stability within the plasma, an extended half-life within tissues, and, consequently, a prolonged therapeutic effect of the ON. Furthermore, these modifications enhance the binding affinity of the ON to the target RNA. The 2′-ribose modifications are incompatible with RNase H activity; thus, they are typically employed for steric blocking or for flanking ON sequences in ASO gapmers [77,78].

Bridged nucleic acids represent a specific class of derivatives characterised by the restriction of the ribose sugar to the 3′-endo conformation, achieved through the formation of a bridge between the 2′- and 4′-carbon atoms. The most commonly utilised variants are locked nucleic acid (LNA) [79], 2′,4′-constrained 2′-O-ethyl (cEt) [80], and, to a lesser extent, 2′-O,4′-C-ethylene-bridged nucleic acid (ENA) [66] (Figure 3c). The incorporation of bridge nucleic acids has been demonstrated to enhance the stability of ONs against nuclease degradation and to improve their hybridisation properties. In the case of LNA, the melting temperature of an ON is increased by 3–8 °C per modified nucleotide. Bridged nucleic acids are incompatible with RNase H-mediated cleavage and have therefore been employed in ASOs aimed at steric blocking [1].

### 4.3. Nitrogenous Base Modifications

The chemical modification of nitrogenous bases enables the regulation of the hybridization properties of ONs. It has been demonstrated that the methylation of pyrimidines to yield naturally occurring 5-methylcytidine or 5-methyluridine/ribothymidine (Figure 3d) results in an increase in the melting temperature of ONs containing them by approximately 0.5 °C for each substitution [50]. Methylated cytosine is a commonly used modification for ASO insertion, as evidenced by its presence in numerous studies [57,59,60]. With regard to the modification of purine bases, 2,6-diaminopurine (Figure 3d) is capable of forming three hydrogen bonds with thymine. This process results in the formation of an analogue of the GC pair. Furthermore, this modification enhances the thermostability of the complementary complex by approximately 1.5 °C per modification. Nevertheless, depurination occurs when ONs are synthesised using acidic conditions to remove dimethoxytrityl protection, which limits the practical application of 2,6-diaminopurine [50].

### 4.4. Alternative Chemistry

Morpholino ONs (PMO), whose structure is based on 6-membered morpholine rings (Figure 5a) and non-ionic phosphorodiamidate bonds (Figure 5b), constitute a distinct group. This modification exhibits remarkable resistance to degradation by a range of hydrolases present in serum and plasma. Due to its uncharged nature, PMO prevents unwanted hybridization with plasma proteins, thereby enhancing the efficacy of tON [81]. To date, several PMO-based drugs aimed at regulating splicing have been approved, as these derivatives are not compatible with the activity of RNase H and RNA interference complex proteins [30,32,82,83]. An alternative approach that has been developed but is not yet widely implemented in clinical practice is the use of peptide nucleic acids (PNA) (Figure 3e) [84]. The principal disadvantage of these chemistries is that both PMO and PNA interact to a limited extent with plasma proteins. As a result, they are rapidly excreted via the urinary system. Furthermore, the potential of tricyclo-DNA (tcDNA) to enhance the stability of RNA-target-ON duplexes by 2.4 °C per modification is also being investigated (Figure 3e) [85].

In conclusion, the achievements in the synthesis of analogues of tONs have enabled the overcoming of nuclease degradation without a notable reduction in therapeutic efficacy, as well as the enhancement of their hybridisation properties. Nevertheless, chemical modification of the sugar–phosphate backbone is insufficient for achieving tissue-specific delivery, efficient penetration of the drug into cells, and its controlled exit from endosomes. Despite the significant advancements in the treatment of various diseases, including viral, oncological, and hereditary ones, there remain unresolved issues regarding the enhancement of the pharmacokinetic properties of tONs while preserving their biological efficacy. The scientific community is engaged in the development of effective and safe approaches to tON delivery. Among these, the attachment of hydrophobic residues to the ON chain, in particular cholesterol, fatty acids, and lipids, as well as alkyl-containing groups, is of particular interest. These modifications contribute to an increase in the efficiency of penetration of such derivatives through the membrane into cells, as well as to the improvement of their pharmacokinetic characteristics [1,3,17].

## 5. Lipophilic Conjugates and Oligonucleotide Derivatives

The delivery potential of ASO and siRNA can be enhanced not only by modifying the structural components of the nucleotide, but also by direct covalent conjugation of various fragments that promote intracellular uptake, targeting the drug to specific cells or tissues, or reducing excretion from the bloodstream [6]. These include lipids [3,23,25], peptides [86,87], aptamers [88,89], antibodies [90,91], and sugars [92,93,94,95]. Bioconjugates are typically smaller in size than other tON delivery vehicles (lipid nanoparticles, liposomes, and exosomes), which results in a favourable biodistribution profile [3]. It is also noteworthy that the toxicity of liposome-, micelle-, and nanoparticle-based delivery systems has been repeatedly demonstrated due to their polycation-induced immunogenicity [96,97]. It has not been reported that lipid-oligonucleotide conjugates exhibit similar immunogenicity. Furthermore, the risk of immunogenicity for modified ASOs is minimal [98,99]. A common approach to bioconjugate design is the selection of conjugated molecules with affinity for cell surface receptors, which subsequently undergo internalisation through receptor-mediated endocytosis. Consequently, the interaction of bioconjugates with receptors can facilitate targeted delivery to specific tissues or cell types within a tissue [1]. Notably, several siRNA-based drugs conjugated to N-acetylgalactosamine (GalNAc) (Figure 6) have been approved for clinical use [31,51,52,53]. These bioconjugates are capable of specifically delivering tONs to the liver with high efficiency [100]. However, the delivery of nucleic acids to other organs is complicated by the need to identify specific receptors present on the surfaces of cells in these organs [101].

The preparation of covalent conjugates of ONs with biomolecules has been the focus of considerable attention. The most prevalent methodologies involve the utilisation of lipid or peptide molecules [22,102]. Nevertheless, the preparation of peptide–oligonucleotide conjugates should be approached with greater caution, as arginine-rich peptides are often found to be highly toxic [103,104]. The utilisation of hydrophobic or lipid compounds, including cholesterol, fatty acids, tocopherol, squalene, and their derivatives, is therefore of particular interest [105,106,107]. The biological effects are demonstrated not only in the organs of excretion, namely the kidney and liver, but also in many others, including the lung, muscle, spleen, and adrenal glands [108]. The benefits of lipid–ON conjugates include favourable biocompatibility and low toxicity [101,109,110]. The lipophilic group enhances the pharmacokinetics and biodistribution of ON in comparison to native unconjugated ONs [111]. It is noteworthy that conjugates of ONs with lipid molecules are characterised by a smaller size in comparison to nanoparticles, and consequently, they are more rapidly transported from the bloodstream to various tissues [112].

The concept of covalently attaching cholesterol to ensure efficient delivery of ONs was first proposed in the late 1990s, and since then, ON cholesterol-containing conjugates for therapeutic applications have been actively developed [113,114,115,116,117,118,119]. Cholesterol interacts with low- and high-density lipoproteins with high efficiency, which facilitates the binding of the resulting complex to cell surface receptors and subsequent penetration [120]. Nevertheless, the delivery area of cholesterol-conjugated ONs remains limited to liver cells and, to a lesser extent, pancreatic cells [108]. Similarly to cholesterol, fatty acids represent an appealing target for bioconjugation, as the variation of the hydrocarbon chain length allows for the alteration of the bioconjugate’s hydrophobicity. The results of active studies of unbranched fatty acids have demonstrated that they may serve as a promising vehicle for the delivery of tON to muscle tissue [121,122,123]. To date, two ASO-based drugs have been approved for the treatment of Duchenne muscular dystrophy [82] and spinal muscular atrophy [58]. However, they are associated with renal toxicity at high doses, and the use of fatty acid conjugates may help to overcome this problem. Vitamin E is a collective term used to describe a group of fat-soluble compounds, including tocopherols and tocotrienols. Of particular interest for chemical bioconjugation is α-tocopherol, which is the natural and most biologically active form of all tocopherols. The structure of α-tocopherol contains a saturated carbon chain, which, due to its hydrophobic properties, increases the permeation ability of ONs through the membrane and enhances their therapeutic effect [17,106,124]. Squalene, a natural triterpene and a precursor of cholesterol in the human body, is also an attractive candidate for the preparation of bioconjugates and has been the subject of extensive research for use in this regard [107,125].

### 5.1. Self-Assembly Features of Amphiphilic Oligonucleotide Derivatives in Aqueous Solutions

Due to their amphiphilic properties, lipid–ON conjugates are capable of self-assembly into micelle-like particles with an average size that can range from 10 to 30 nm. This variability is dependent on the length of the oligonucleotide and the length of the alkyl groups [109,126,127,128,129,130]. In aqueous solutions, these particles consist of a hydrophobic core surrounded by a hydrophilic ON corona [126,129]. For biomedical applications, micellar structures are of particular interest due to their small size, favourable biocompatibility, and capacity to facilitate intracellular accumulation of tONs [109,126,128,131]. The efficacy of such drugs can be monitored by controlling their size and morphology [126,132]. A number of amphiphilic ON derivatives have been synthesised and characterised in [126]. In aqueous solutions, amphiphilic ON molecules comprising a hydrophilic ON covalently linked to hydrophobic diacillipid tails (Figure 7) spontaneously self-assemble into monodisperse three-dimensional micellar nanostructures with a lipid core and an ON corona. Experimental evidence indicates that these types of micelles exhibit excellent thermal stability, and their size can be precisely controlled by varying the length of the ON sequence. Upon increasing the length of the oligonucleotide from five monomeric units to fifty, a notable increase in micelle size was observed, from 7.8 to 36.4 nm. Furthermore, the molecular recognition properties are retained within the micellar system, allowing ONs to hybridise with complementary sequences within the micelles while maintaining their structural integrity. It is noteworthy that upon interaction with cell membranes, highly charged micelles are capable of disintegrating and incorporating into the cell membrane, thereby completing the internalisation process through endocytosis. According to flow cytometry data, micelles with a size of 7.8 nm have been observed to result in a 601-fold increase in cell fluorescence intensity relative to untreated cells. However, when cells are treated with 36.4 nm micelles, this enhancement is reduced to 85 times. Therefore, the kinetics of this internalisation process have been demonstrated to be size-dependent.

In a related study [127], it was demonstrated that the size and morphology of nucleic acid-based supramolecular structures can be regulated by means of hybridisation interactions. ON–lipid conjugates (Figure 8) undergo self-assembly to form spherical vesicles (with a diameter exceeding 100 nm), which are capable of undergoing a reversible change in morphology to form small spherical micelles of approximately 20–25 nm in diameter. This transformation occurs through cycles of hybridisation and chain displacement.

Karaki et al. employed lipid bioconjugation and subsequent micelle formation for the purpose of developing antisense ONs against translationally controlled tumour protein (TCTP) mRNA [109]. The authors demonstrated that ASOs modified with 15-carbon alkyl groups (Figure 9) self-assemble into spherical objects with diameters of ~11 nm, exhibiting enhanced penetration and efficacy in inhibiting TCTP expression in the absence of transfection agents, both in vitro and in vivo. Transfection of the bioconjugates resulted in rapid and prolonged internalisation via macropinocytosis, accompanied by the suppression of TCTP and a significant reduction in cell viability. Furthermore, the derivatives were demonstrated to result in delayed tumour progression in CRPC xenograft models, without any significant toxic effects. The same lipid conjugates have been demonstrated to be effective for the delivery of ONs not only into eukaryotic cells, but also into prokaryotic cells. In [128], the strong antimicrobial potential of these ON derivatives was demonstrated by targeting β-lactamase activity on both clinical and laboratory strains. These results support the concept that the self-delivery of ON sequences via lipid conjugation can be extended not only to ASOs, siRNAs, and aptamers, but also to antimicrobials. This represents a promising approach for combating antibiotic resistance in bacteria.

In [131], a group of authors conducted a comprehensive analysis of the impact of length, degree of unsaturation, and the number of fatty acid (FA) residues (myristic acid, docosahexaenoic acid, eicosapentaenoic acid) conjugated to siRNA (Figure 10). The siRNAs conjugated with a single FA residue did not undergo aggregation, and their average diameter was 2.5 nm, which was similar to that of unconjugated siRNAs. In contrast, siRNA conjugated with two and three FAs did self-assemble into small aggregates and micelles. The mean particle size was considerably larger for conjugates with three residues in comparison to those with two residues (with diameters of 11 and 4 nm, respectively). The degree of unsaturation and carbon chain length of the fatty acid did not have a significant impact on the size of the micelles formed. The results demonstrated that fatty acid-conjugated siRNAs exhibited gene accumulation and repression in extrahepatic tissues, indicating the potential for extending the therapeutic applications of siRNAs beyond the liver. The lipid-conjugated siRNAs with three FA residues demonstrated preferential retention at the injection site, with minimal systemic exposure. In contrast, the mono-conjugated siRNAs exhibited rapid release into circulation, resulting in a preferential accumulation in the kidneys. The siRNAs conjugated to two fatty acid residues displayed intermediate behaviour, with the majority accumulating in the liver and exhibiting functional distribution in the lung, heart, and adipose tissue. These findings may provide a foundation for the development of chemical modification strategies to enhance the delivery of lipophilic siRNAs to extrahepatic tissues.

Kusznir et al. conducted a systematic study with a similar methodology to investigate the propensity for self-assembly of a number of ASO gapmers modified by FA via an aminohexyl linker specific to the MALAT1 gene. To this end, a library of ASOs was synthesised and covalently modified with saturated FAs of varying length, branching, and 5′/3′-attachment [132]. The authors demonstrated that ASOs conjugated to fatty acids longer than C16 exhibited an increasing propensity to form self-organising vesicular structures. However, it was observed that the formation of self-organising structures was accompanied by an increase in stability, which was found to be proportional to the length of the fatty acid chain. To illustrate, FA chains with lengths shorter than C24 readily formed self-organising structures comprising 2 (C16), 6 (C22, bis-C12), and 12 (C24) monomers. In contrast, ASO modification with dipalmitic acid (C32) resulted in the formation of a robust hexameric complex. The results demonstrate that the nature of the mono- and multimeric structures of hydrophobic-modified ASO is determined by the hydrophobic effect. Therefore, the formation of dispersed structures is a direct consequence of the chain length of the fatty acid. This offers the potential for the concept of hydrophobic modification to be employed in order to influence the pharmacokinetics and biodistribution of ASO.

In conclusion, the amphiphilic nature of ON–lipid conjugates implies a propensity to form micelles and other aggregates, including vesicles and more complex self-assembling structures, in aqueous solutions. The structures typically exhibit a spherical shape, with notable variations in size and morphology depending on the experimental conditions, the nature of the lipophilic group, and the length of the ON sequence.

### 5.2. Interaction of Lipid-Containing Oligonucleotides with Serum Albumin

The enhancement of pharmacokinetic characteristics is typically ascribed to the interaction of lipid–ON conjugates with serum proteins [120,124]. It is also postulated that their association with one or more serum proteins (such as albumin) is involved in their cellular uptake mechanisms and thus strongly influences their biodistribution. Among the variety of hydrophobic molecules, those with a higher affinity for binding to HSA are of particular interest, since its molecule contains at least seven fatty acid binding sites [133]. HSA represents the predominant protein component of human serum, accounting for approximately 60% of the total protein content. Due to its low immunogenicity, long half-life, and capacity to accumulate in tissues undergoing oncotransformation and inflammation, as well as the presence of binding sites for various ligands in its structure, this protein is the subject of numerous studies worldwide aimed at the development of delivery systems for a wide range of therapeutic and diagnostic agents [134,135,136,137,138]. It is established that HSA has a significant impact on the pharmacokinetics and pharmacodynamics of drugs [139,140,141]. Furthermore, the binding of a chemical compound to this protein can result in an increase in its solubility in plasma or a reduction in its toxicity [142,143,144]. The enhanced permeability and retention (EPR) effect observed in tumours and their surrounding tissues, coupled with the presence of specific receptors (Gp60, SPARC, FcRn) that are overexpressed in certain cancers, makes HSA-binding constructs an attractive tool for cancer diagnostics, therapy, and theranostics.

The interaction of tONs with serum and cellular proteins is a determining factor in their pharmacokinetic properties (transport to and distribution in target tissues) and pharmacodynamic properties (binding to target RNA), which in turn determine their final biological effect. Consequently, the impact of non-covalent HSA binding on ON biodistribution following administration has been a subject of intensive investigation over recent decades. The stability of non-covalent complexes of albumin with various amphiphilic oligonucleotide derivatives is summarised in Table 2.

The earliest documented attempts to conjugate ASOs with small molecules, particularly ibuprofen (Figure 11), to regulate interactions with HSA were reported in 2002 [145]. It was demonstrated that 2′-deoxy and 2′-MOE ONs exhibited minimal binding affinity to HSA (Kd > 400 μM), whereas PS derivatives demonstrated a significantly higher affinity (Kd ≈ 10^−6^ M). Furthermore, the conjugation of the 3′-end of 2′-deoxy and 2′-MOE ONs with ibuprofen resulted in an enhanced binding affinity comparable to that observed for PS derivatives. It was, therefore, concluded that the administration of ibuprofen is capable of influencing the distribution of ONs in tissues. Furthermore, given that the observed binding capacity is in the micromolar range, it can be concluded that the release of the ON from the complex with the protein should not be an issue. Given that ibuprofen is situated at the 3′-end, it can be concluded that the hybridisation properties of ASO to the target should remain unaltered, while the nuclease resistance should be enhanced.

As previously stated, research is ongoing into cholesterol-conjugated ONs and the mechanisms of their enhanced efficacy in comparison to their unconjugated analogues. Antisense ONs containing PS modifications and cholesterol at the 3′-end demonstrate high affinity for serum proteins, particularly HSA, as well as high- and low-density lipoproteins. The dissociation constant (Kd) of the ON–protein complexes obtained in this study varies from 227 nM to 0.6 nM, respectively [120,124]. The conjugation of cholesterol results in high accumulation of phosphorothioate oligodeoxynucleotides in various types of liver cells. It is noteworthy that cholesterol at the 5′-end of the ON (Figure 12) in combination with PS modifications also contributes to the ASO effect in muscle tissue. However, compared to α-tocopherol or palmitic acid conjugates, a cholesterol-containing ON has a more pronounced toxic effect at high doses [124].

In a related study, the transport properties of HSA were employed to regulate the plasma half-life, accumulation in the liver, and biological activity of siRNA depending on the amount of cholesterol modifications [148]. The authors demonstrated that recombinant HSA exhibited robust and selective binding to cholesteryl-modified siRNA, with a Kd of approximately 10⁻⁷ M, which exhibited an inverse correlation with the amount of cholesterol. The resulting complexes exhibited reduced susceptibility to nuclease degradation and also demonstrated an inhibitory effect on the induction of TNF-α production by human peripheral blood mononuclear cells. Following in vitro screening of GFP silencing, the optimal siRNAs were identified as those containing sugar–phosphate backbone modifications (LNA, 2′-F, 2′-OMe or PS) in addition to cholesterol residues. The introduction of one or two cholesterols into the siRNA resulted in a significant increase in the half-life of the ON in blood in mice, from 12 min (for unmodified siRNA) to 45 or 71 min, respectively. Biodistribution experiments demonstrated that the accumulation of siRNA with two cholesterols in the liver was increased, which correlated with a 28% increase in biological activity compared with a 4% increase for native siRNA.

Similarly, fatty acids and tocopherol represent promising targets for ON functionalisation. In a comparative study, Østergaard et al. examined the efficacy of three distinct bioconjugates of PS-modified ASOs in binding to cholesterol, tocopherol, and palmitic acid (Figure 13) [124]. The conjugate with α-tocopherol exhibited the highest affinity for HSA (Kd = 24 nM), while the lowest affinity was observed for cholesterol (Kd = 227 nM). The dissociation constant for the complex of the ON conjugate with palmitic acid and HSA was determined to be 218 nM. In addition, the palmitate-conjugated compound demonstrated superior efficacy in inhibiting myotonin–protein kinase synthesis in the skeletal muscle and heart tissue in rat models. Nevertheless, the ASO with tocopherol demonstrated superior efficacy in the monkey model.

The ability to regulate the hydrophobic properties of ON derivatives by modifying the length of the hydrocarbon chain represents a significant advantage of fatty acids. In [149], a technology is presented that also employs the natural transport functions of HSA to enhance the pharmacokinetic properties of ASO. The self-assembly of fatty acid-modified ONs (Figure 14) and HSA into supramolecular structures was demonstrated to result in favourable pharmacokinetics. The interaction with albumin is contingent upon the length of the carbon skeleton of the fatty acid (palmitic (C16) or myristic acid (C14)), the number of residues and their position in the ASO sequence, and the presence of PS modifications. The binding of the investigated derivatives containing two palmitic acid residues to HSA was found to correlate with an increase in blood circulation time in mice, from 23 min to 49 min for PO-ONs and from 28 min to 66 min for PS-modified ASO. Furthermore, a shift towards enhanced biodistribution was observed for PS in comparison to PO ASO. The incorporation of two palmitic acid residues resulted in the ON distribution resembling that of HSA. Consequently, this study presents an alternative approach based on endogenous assembly of ASO/HSA constructs to prolong circulation and modulate ASO biodistribution, as well as to obtain tunable pharmacokinetics through ON sequence design.

In a follow-up to the promising results obtained in [149], Prakash’s group conducted a comprehensive investigation into the activity and binding of ASO conjugates (Figure 15) to blood plasma proteins. This involved the examination of fatty acids differing in chain length, degree of unsaturation, and double bond conformation (cis/trans isomers) [121]. The affinity of ASO for plasma proteins increased with increasing fatty acid chain length, with the highest binding affinity observed at lengths between 16 and 18 carbons. The degree of unsaturation and conformation of the double bond did not appear to affect the binding of ASO fatty acid conjugates to proteins. The activity of the conjugates was found to correlate with their affinity for albumin, with the strongest binding to albumin demonstrating the greatest increase in activity in muscle. The results indicate that the optimal fatty acids for improving functional ASO uptake in skeletal and cardiac muscle are palmitic acid and oleic acid. Subsequently, a team of researchers demonstrated that the administration of palmitic acid-conjugated ASOs to mice resulted in the rapid and significant accumulation of these compounds in the interstitium of muscle tissue [150]. This was followed by a relatively rapid clearance, with only a slight increase in intracellular accumulation observed in myocytes. A model has been proposed which suggests that increased affinity for albumin and other plasma proteins due to lipid conjugation facilitates ASO transport across endothelial barriers into the tissue interstitium. However, this also permits the transport of ASO from the interstitium into the lymphatic system and back into the bloodstream. The overall effect is a minimal increase in tissue accumulation (approximately twofold) and a comparable rise in ASO activity. To confirm this assumption, it was demonstrated that the activity of fatty acid-conjugated ASOs was reduced in two mouse models with defects in endothelial macromolecule transport as a result of caveolin-1 knockout and FcRn knockout. Nevertheless, this research has established a foundation for the advancement of more efficacious therapeutic ASOs, with the potential to facilitate the development of treatments for muscle tissue-related diseases.

In a recent publication [132], the authors investigated not only the self-assembly propensity of a series of FA-modified ASO gapmers, but also the impact of non-covalent binding to HSA on the formation of supramolecular structures. The ASOs were modified covalently with saturated fatty acids of varying lengths, branching, and 5′/3′-attachment. The C16–C24 series demonstrated the capacity to interact with mouse and human HSA via their fatty acid chains, forming stable adducts. This interaction exhibited a nearly linear correlation between the hydrophobicity of the FA-ASO and the binding strength to mouse albumin. The incubation of the conjugates, in which the FA chain length was shorter than C24, with albumin resulted in the disruption of the supramolecular structures that had formed. The formation of FA-ASO/albumin complexes was observed, with a 2:1 stoichiometry and binding affinity in the low micromolar range. The hexameric complex formed by ASO modified with dipalmitic acid (C32) was observed to remain stable upon incubation with albumin at conditions above the critical aggregation concentration. In light of these findings, it becomes evident that the concept of hydrophobic modification can be employed to influence the pharmacokinetics and biodistribution of ASO in two distinct ways: firstly through the binding of FA-ASO to albumin and secondly through the self-assembly process, which ultimately leads to the formation of supramolecular structures. Both concepts offer the potential to influence the biodistribution, receptor interaction, cell penetration mechanism, and in vivo pharmacokinetic/pharmacodynamic properties, thereby facilitating delivery to extrahepatic tissues at a concentration sufficient to treat the disease.

Polymer-based cationic reagents, such as Lipofectamine 2000 or jetPEI, which form liposomes or nanoparticles, are commonly employed for the purpose of nucleic acid transfection, both in vitro and in vivo. A research group attempted to develop an alternative to jetPEI for siRNA delivery using HSA as a carrier [151]. Two stearic acid residues were conjugated via a linker to the passenger chain of the siRNA (Figure 16), resulting in the rapid binding of the siRNA to albumin in situ. The resulting conjugate demonstrated a 5.6-fold increase in half-life, an 8.6-fold improvement in bioavailability, and reduced accumulation in the kidney when compared to unmodified siRNA. In comparison to the most prevalent commercial siRNA nanocarrier in vivo, jetPEI, the lipid conjugate siRNA demonstrated a 19-fold accumulation in the tumour and a 46-fold increased uptake in tumour cells. The siRNA modified with stearic acid residues demonstrated rapid penetration into tumour tissue, with uptake in 99% of tumour cells achieved 30 min after intravenous administration, in comparison to 60% for jetPEI. The enhanced pharmacokinetic properties of the siRNA contributed to a notable reduction in tumour gene expression within seven days following the administration of two intravenous doses. The presented data demonstrate considerable potential for in situ albumin targeting in the development of siRNA-based, carrier-free cancer therapies.

An additional ligand that can be employed to enhance the hydrophobicity of ONs is an alkyl group. The conjugation of dendrite-like alkyl chains to the 5′-end of deoxyribonucleotide has been described previously in detail in [146,152] (Figure 17). The resulting amphiphilic conjugate exhibits a high degree of binding to HSA under physiological conditions, with a dissociation constant (Kd) in the nanomolar range. This results in an increase in their half-life in serum. Furthermore, it has been observed that ONs exhibit reduced degradation under physiological conditions when exposed to nucleases, as well as reduced uptake and degradation by immune cells (macrophages). In other words, albumin binding can be employed to reduce non-specific uptake (by healthy and immune cells) of ON drugs, thereby potentially enhancing ASO delivery to target cells. The evidence presented above provides further support for the hypothesis that the incorporation of fragments into ONs that facilitate binding to HSA can be employed as a strategy to regulate their biodistribution.

The conjugation of ONs with three non-nucleotide dodecyl-containing moieties (Figure 18) has been observed to enhance their affinity for HSA in comparison to that of free ONs [129,153]. The authors observed the formation of albumin–ON complexes with a stoichiometry of approximately 1:1.25 ± 0.25 under conditions that closely resemble physiological conditions. Atomic force microscopy revealed that the interaction of human HSA with a duplex of complementary dodecyl-containing ONs resulted in the formation of ring-shaped associates with a diameter of 165.5 ± 94.3 nm and a height of 28.9 ± 16.9 nm. In contrast, the interaction with polydesoxyadenylic acid and dodecyl-containing oligotimidylate led to the formation of supramolecular associates with a diameter of 315.4 ± 70.9 nm and a height of 188.3 ± 43.7 nm, respectively. The conjugate and its duplex were efficiently internalised and accumulated in the cytoplasm of HepG2 cells without the use of a transfection agent. The addition of HSA or foetal bovine serum to the medium was demonstrated to reduce the efficiency of ON uptake by 22.5–36%, while not completely inhibiting cell penetration. The data obtained allow us to consider dodecyl-containing ONs and albumin as potential components of the self-organising systems created for the development of unique theranostic agents with targeted action.

As previously highlighted, the inter-nucleotide phosphate represents a potential site for the introduction of modifications aimed at enhancing the nuclease resistance of tONs. The combination of the advantages of such nucleic acid derivatives with rational, scalable synthesis is achieved by the introduction of dodecyl fragments via inter-nucleotide phosphate in ONs. The work [130] is dedicated to the characteristics of non-covalent interactions with human HSA and their impact on the formation of micelle-like structures of triazinyl phosphoramidate ONs (Figure 19). The study demonstrated that the introduction of a triazinyl phosphoramidate modification bearing two dodecyl groups in the 3′-terminal region of the ON sequence had a negligible effect on its affinity for the complementary sequence. Dynamic light scattering demonstrated that the examined amphiphilic ONs were capable of self-organising into micelle-like particles with diameters ranging from 8 to 15 nm. The formation of stable complexes between ONs with dodecyl groups and human HSA was observed with a dissociation constant of approximately 10^−6^ M. However, the micelles of the ONs disintegrated upon binding to albumin. The data presented here, together with the ability of dodecyl-containing derivatives to penetrate a range of cells, including HEK293 and T98G, suggests that the ONs under investigation could be developed into a platform for therapeutic drugs with targeted action.

A recent research paper [147] presented a general approach to regulate the interactions between ON and HSA by modifying the hydrophobic properties of ON. However, in contrast to previous work, a library of DNA aptamers with varying hydrophobic F base content was designed and synthesised (Figure 20). In vitro experiments demonstrated that the incorporation of two F bases at both ends of the aptamer enhanced its nuclease resistance without compromising its biological activity. Furthermore, the binding capacity towards albumin exhibited a notable increase, with a dissociation constant (Kd) ranging from 100 nM to 1 µM. In vivo experiments demonstrated that the aptamer–albumin complex was immediately formed following injection, and that the aptamer’s circulation time was significantly prolonged due to enhanced resistance to nuclease degradation and delayed excretion through the kidneys. The incorporation of F-bases represents a general approach to regulating the albumin-binding capacity of aptamers and enhancing their stability in vivo, thereby prolonging the circulation time of aptamer therapeutics.

In conclusion, the incorporation of hydrophobic fragments into ONs enhances their interaction with cell membranes, facilitates their internalisation, and influences their distribution in vivo. Once introduced into biological fluids, these modifications facilitate the binding of various serum proteins, which in turn play a crucial role in determining the distribution of ONs within the body. HSA, the most abundant circulating blood protein, represents an attractive candidate for study due to its previous utilisation to enhance the therapeutic effects of various therapeutic agents. The results of numerous studies demonstrate that strong binding to HSA can be employed as a strategy to reduce enzyme-induced degradation of ONs under physiological conditions (nucleases), to reduce uptake and degradation by immune cells (macrophages), and to prevent non-specific cellular uptake. The incorporation of protein-binding groups into ONs represents a strategy for controlling the biodistribution of these agents within the body and overcoming the numerous barriers they encounter, including stability, immunogenicity, and off-target effects.

## 6. Conclusions

The advances in genetics and nucleic acid chemistry have provided the foundation for the development of novel therapeutic drugs that are capable of selectively affecting molecular targets. The primary challenge to the widespread clinical utilisation of tONs is the issue of delivery into cells. This literature review considers one of the strategies used to date to improve the pharmacodynamic and pharmacokinetic characteristics of tONs: bioconjugation with lipid molecules. It is evident that lipid conjugates accumulate in the secretory organ, the liver. However, their detection in other tissues, including the spleen, kidney, and even skeletal muscle tissue, opens the way for the development of new methods of delivery to extrahepatic tissues. Understanding the effects of non-covalent binding of HSA to amphiphilic ON derivatives on their self-assembly into micelle-like structures, biodistribution in the body, and therapeutic effects cannot be underestimated. Interactions with this protein may provide new opportunities and properties for tONs, including increased stability, reduced immune response, and controlled biodistribution. The desired outcome of numerous studies is to optimise the design of ON sequences and obtain highly stable and bioactive therapeutic drugs.

## Figures and Tables

**Figure 1 pharmaceutics-16-01447-f001:**
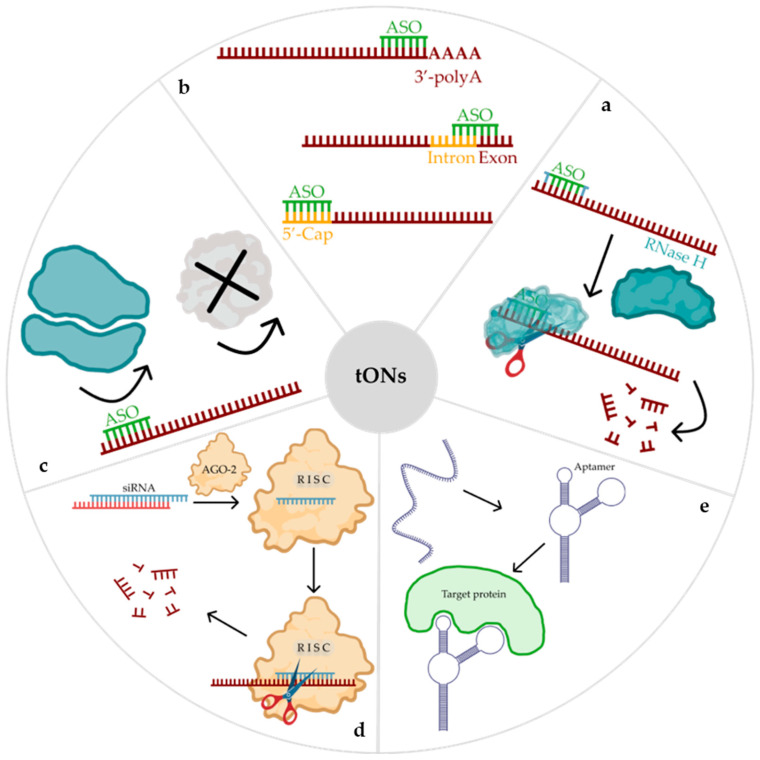
Main therapeutic oligonucleotide (tON) mechanisms of action: (**a**)—RNase H-dependent mechanism of mRNA degradation; (**b**)—regulation of mRNA maturation process (regulation of splicing, inhibition of capping, inhibition of polyadenylation); (**c**)—blocking the landing site of the ribosome, resulting in the inhibition of translation; (**d**)—RNA interference; (**e**)—altering protein conformation and functions.

**Figure 2 pharmaceutics-16-01447-f002:**
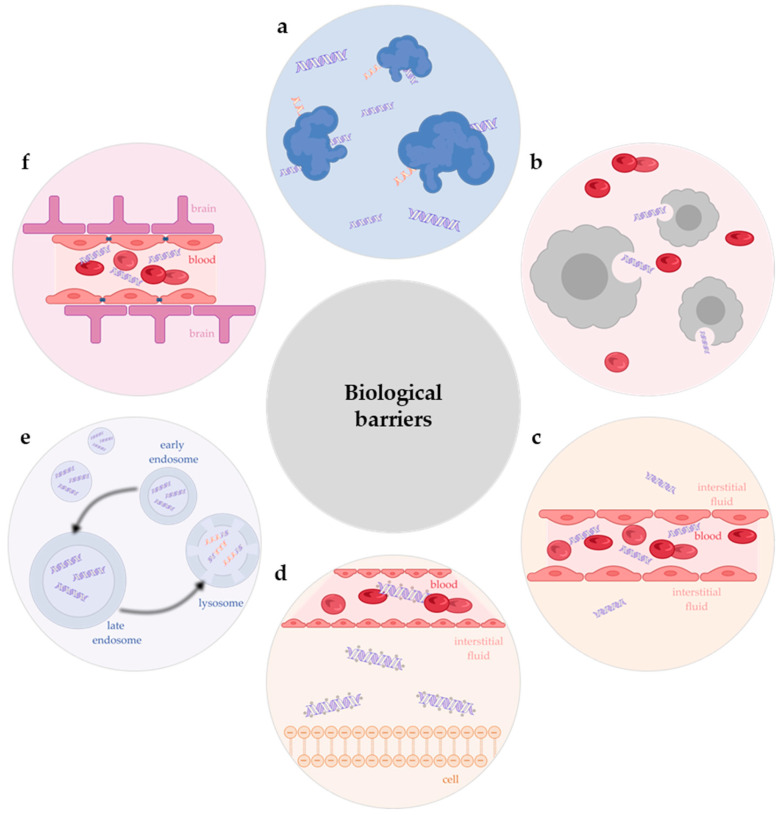
Biological barriers that influence the pharmacokinetic characteristics of oligonucleotide-based drugs: (**a**)—nuclease degradation; (**b**)—reticuloendothelial system; (**c**)—blocking the lateral movement of oligonucleotide from the vessel lumen into the interstitial fluid; (**d**)—low efficiency of cell penetration; (**e**)—endo-lysosomal entrapment; (**f**)—blood–brain barrier.

**Figure 3 pharmaceutics-16-01447-f003:**
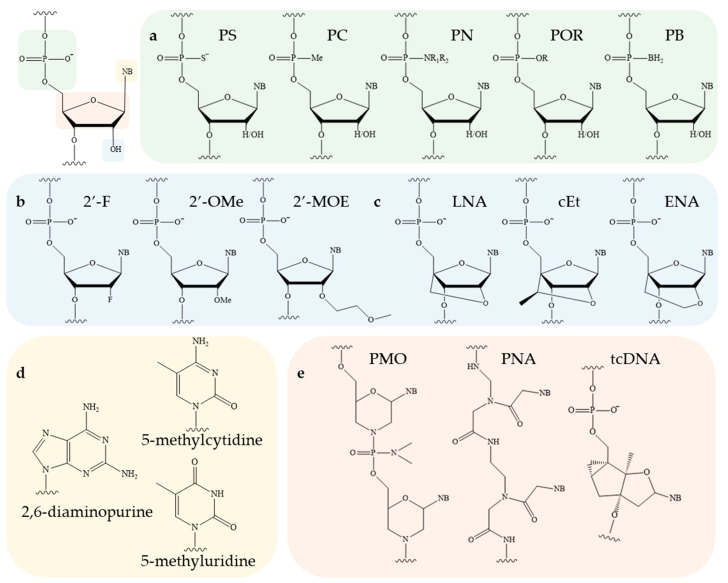
The structure of RNA nucleotide and potential sites for its chemical modification: (**a**)—modifications of inter-nucleotide phosphate; (**b**,**c**)—modifications of sugar backbone; (**d**)—modifications of nitrogenous bases; (**e**)—alternative chemistry (NB—nitrogenous base).

**Figure 4 pharmaceutics-16-01447-f004:**
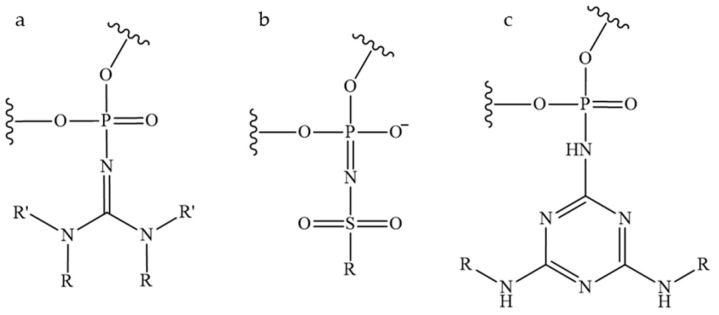
New classes of phosphate-modified oligonucleotides: (**a**)—phosphoryl guanidine derivatives; (**b**)—sulfonyl phosphoramidate derivatives; (**c**)—triazinyl phosphoramidate derivatives.

**Figure 5 pharmaceutics-16-01447-f005:**
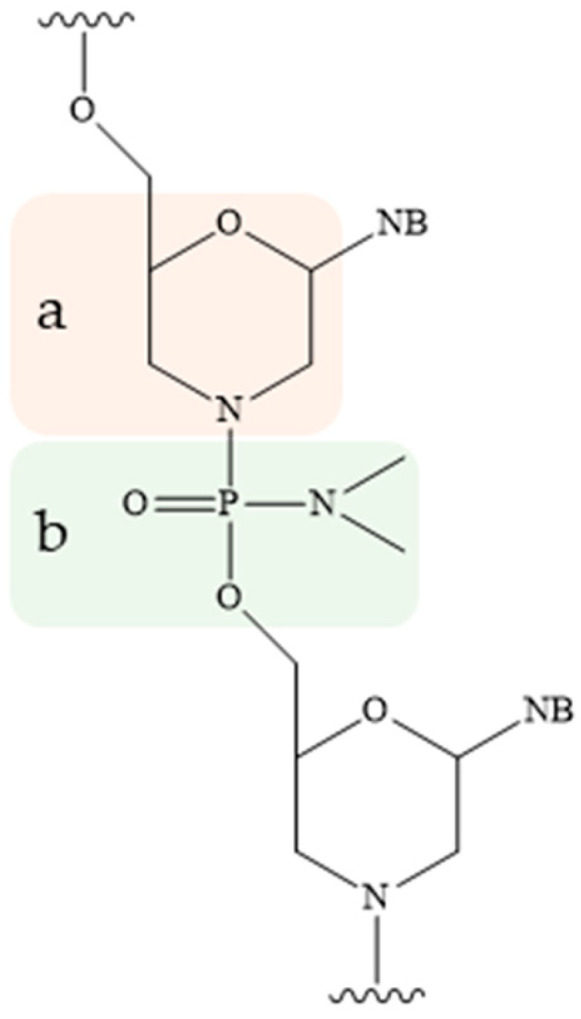
Chemical structure of morpholino oligonucleotides: (**a**)—6-membered morpholine rings, (**b**)—non-ionic phosphorodiamidate bond (NB—nitrogenous base).

**Figure 6 pharmaceutics-16-01447-f006:**
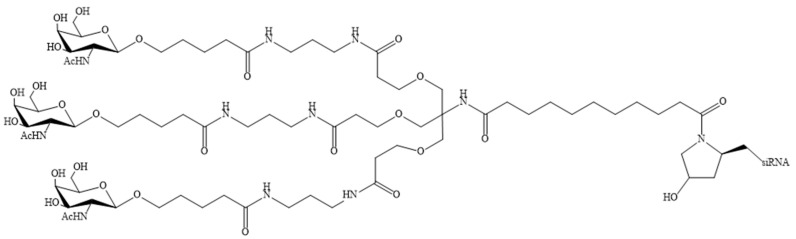
N-acetylgalactosamine (GalNAc) fragment conjugated to siRNA.

**Figure 7 pharmaceutics-16-01447-f007:**
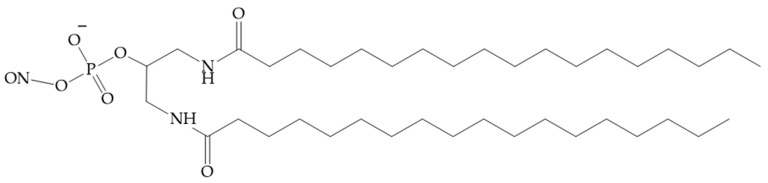
Structural formula of a lipid moiety containing two C18 alkyl groups, which are covalently linked to the 5′-end of the ON (ON—oligonucleotide sequences containing five to fifty nucleotide units).

**Figure 8 pharmaceutics-16-01447-f008:**
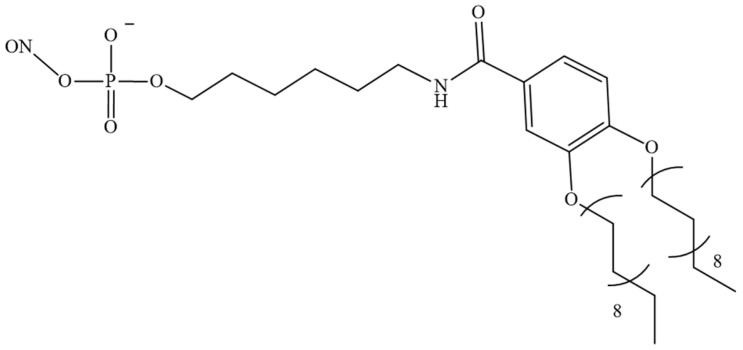
An ON–lipid conjugate comprising two 18-carbon alkyl groups, which are covalently linked to the 5′-end of the ON (ON—oligonucleotide sequence).

**Figure 9 pharmaceutics-16-01447-f009:**
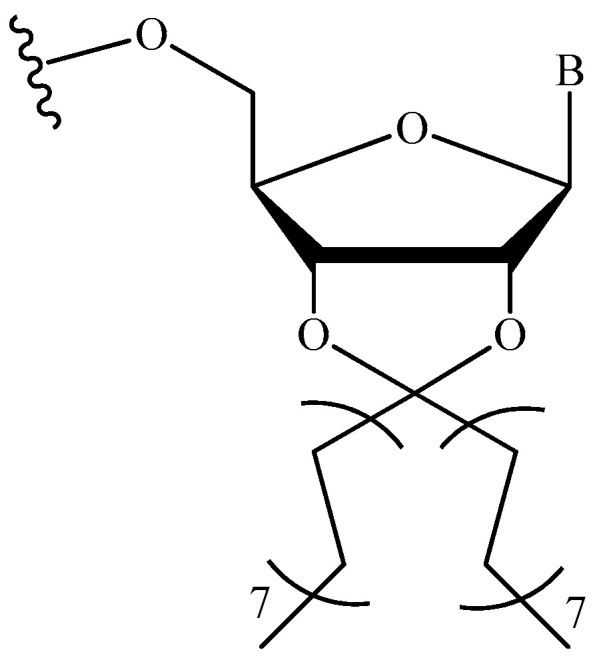
The chemical structure of the modified nucleotide linkage employed in [109,128] (B—nitrogenous base).

**Figure 10 pharmaceutics-16-01447-f010:**
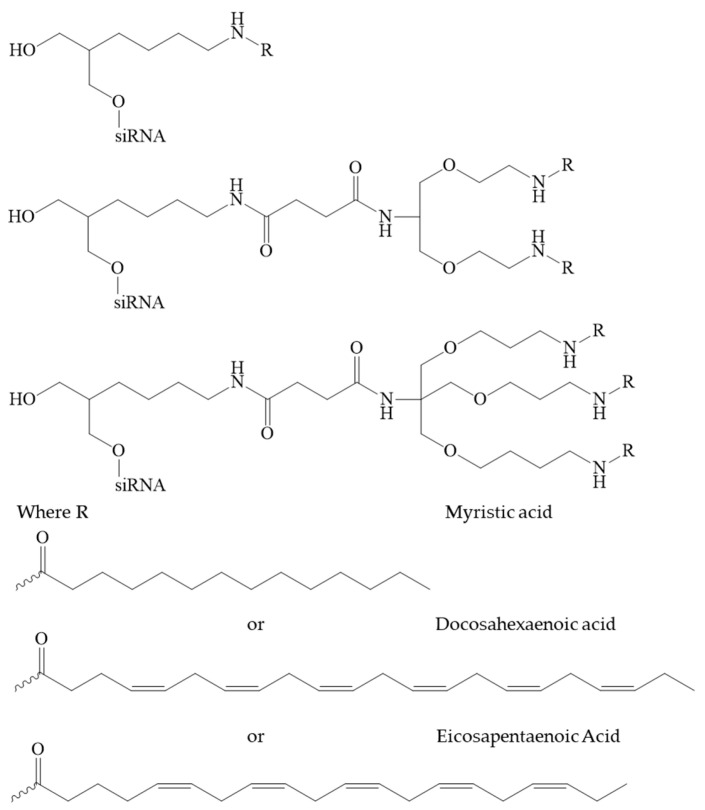
The chemical structures of the siRNA conjugates in [131].

**Figure 11 pharmaceutics-16-01447-f011:**
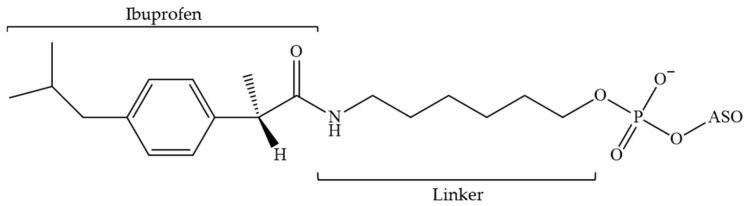
The chemical structure of the 3′-terminal link has been modified at the 3′-position by (S)-(+)-ibuprofen (ASO—antisense oligonucleotide).

**Figure 12 pharmaceutics-16-01447-f012:**
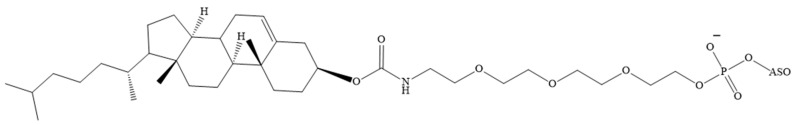
The structure of the cholesterol fragment conjugated to PS ASO (ASO—antisense oligonucleotide).

**Figure 13 pharmaceutics-16-01447-f013:**
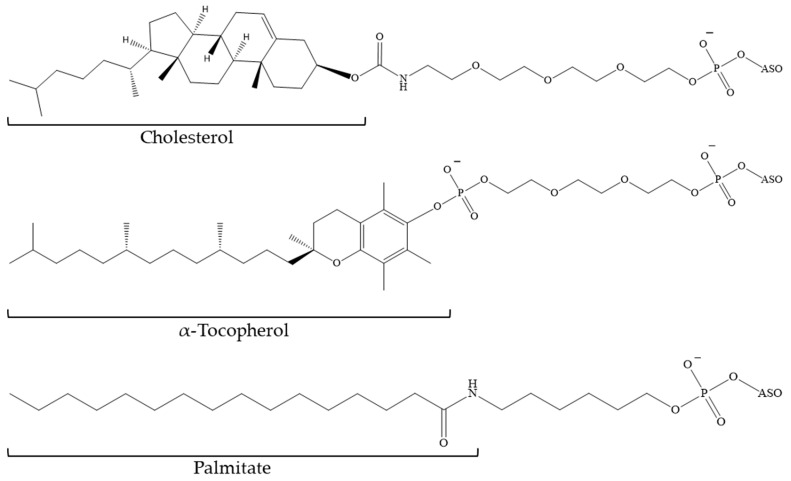
The structures of the fragments conjugated to PS ASO (ASO—antisense oligonucleotide).

**Figure 14 pharmaceutics-16-01447-f014:**
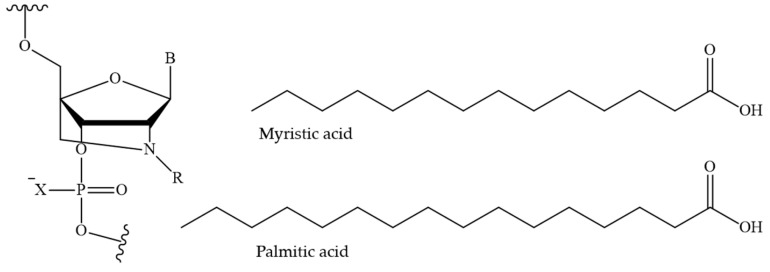
The chemical structure of the N_2_′-functionalised amino-LNA monomer (R—myristic acid or palmitic acid residue; B—nitrogenous base; X—O or S atom).

**Figure 15 pharmaceutics-16-01447-f015:**
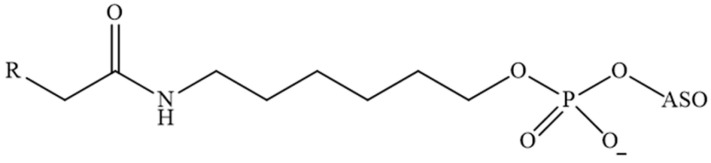
The chemical structure of the 5′-terminal link (ASO—antisense oligonucleotide; R—fatty acid residue).

**Figure 16 pharmaceutics-16-01447-f016:**
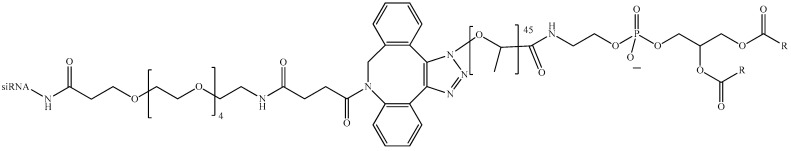
siRNA conjugated to stearic acid (siRNA—small interfering RNA; R—residue C_17_H_35_).

**Figure 17 pharmaceutics-16-01447-f017:**
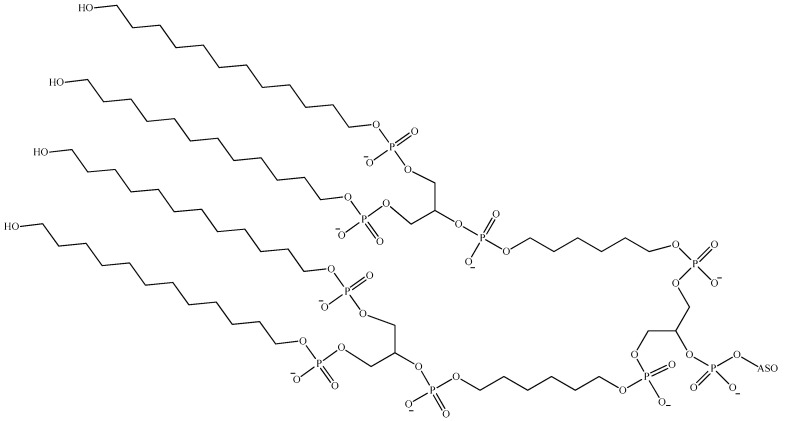
The chemical structure of the 5′-terminal nucleotide linkage (ASO—antisense oligonucleotide).

**Figure 18 pharmaceutics-16-01447-f018:**
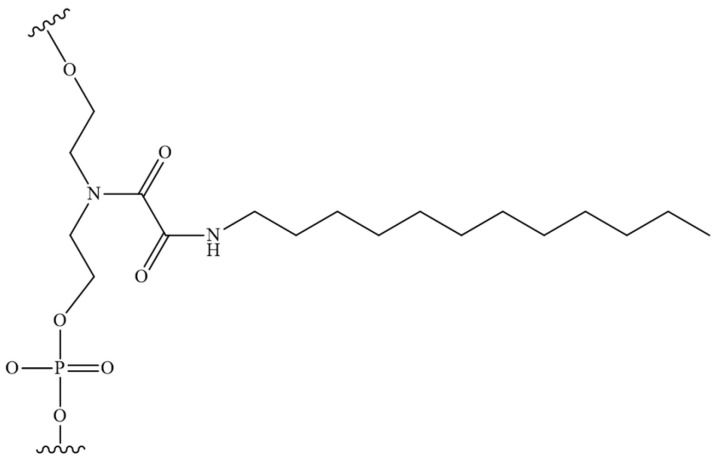
The chemical structure of a non-nucleotide dodecyl-containing link within an oligonucleotide.

**Figure 19 pharmaceutics-16-01447-f019:**
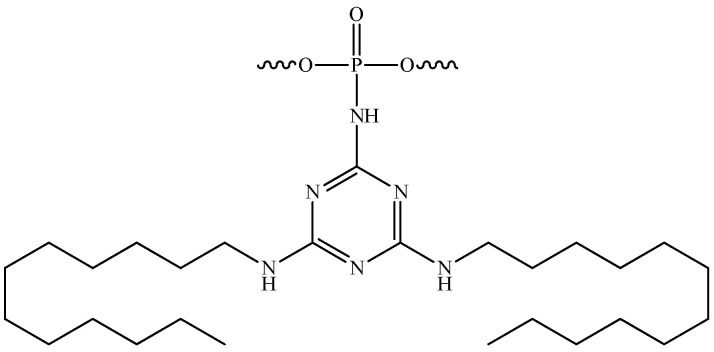
The chemical structure of triazinyl phosphoramidate modification functionalized with two dodecyl groups.

**Figure 20 pharmaceutics-16-01447-f020:**
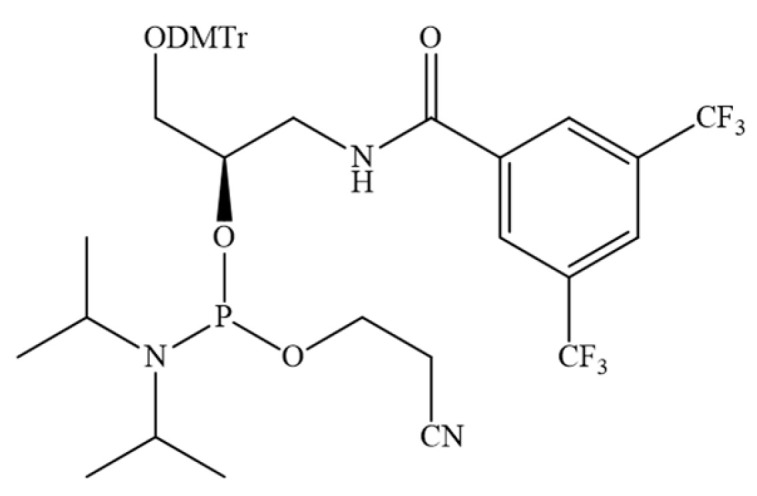
The chemical structure of the phosphoramidite used to insert the F base into the aptamer sequence (DMTr—dimethoxytriphenylmethyl).

**Table 1 pharmaceutics-16-01447-t001:** Half-life times of oligonucleotides in the presence of 50% fetal bovine serum.

Oligonucleotide	Modification	Half-Life Time (τ½)
agtctcgacttgctacc	deoxy/PO	10 min
a_S_g_S_t_S_c_S_t_S_c_S_g_S_a_S_c_S_t_S_t_S_g_S_c_S_t_S_a_S_c_S_c	deoxy/PS	10 h
a*g*t*c*t*c*g*a*c*t*t*g*c*t*a*c*c	deoxy/PG	>21 days
AGUCUCGACUUGCUACC	2′-OMe/PO	70 min
A_S_G_S_U_S_C_S_U_S_C_S_G_S_A_S_C_S_U_S_U_S_G_S_C_S_U_S_A_S_C_S_C	2′-OMe/PS	>24 h
A*G*U*C*U*C*G*A*C*U*U*G*C*U*A*C*C	2′-OMe/PG	>21 days

**Table 2 pharmaceutics-16-01447-t002:** K_d_ of the complexes of albumin with modified oligonucleotides.

Modification	Site of Modification	K_d_, μM	References
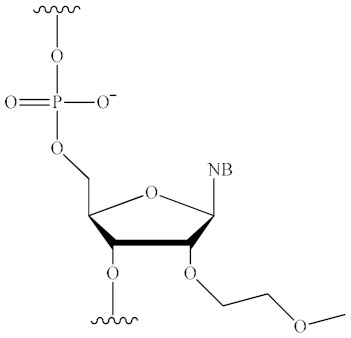	sugar backbone, 2′-OH	>400	[145]
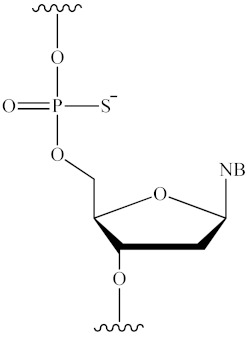	inter-nucleotide phosphate	4–7	[145]
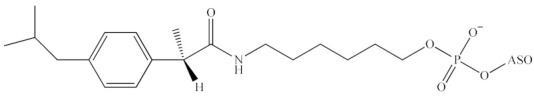	3′-end of ON	8–12	[145]
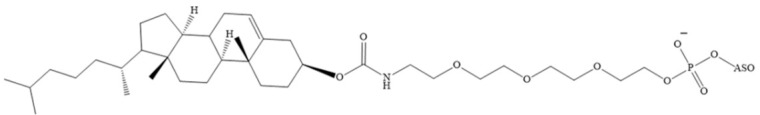	3′-end of ON	0.227	[124]
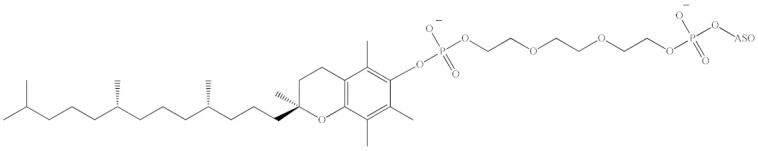	3′-end of ON	0.024	[124]
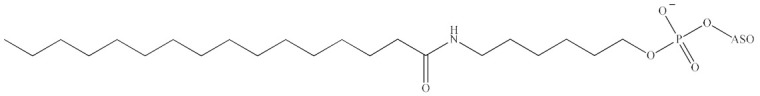	3′-end of ON	0.218	[124]
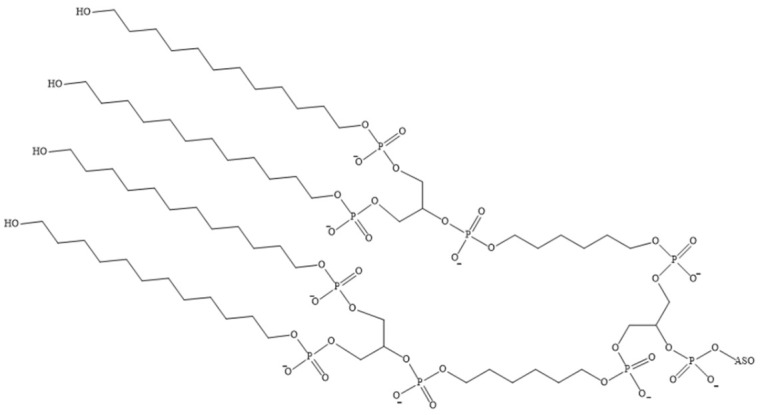	3′-end of ON	0.041	[146]
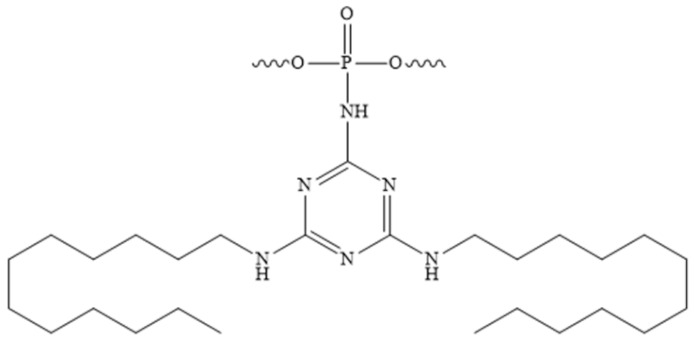	inter-nucleotide phosphate	1.2–1.5	[130]
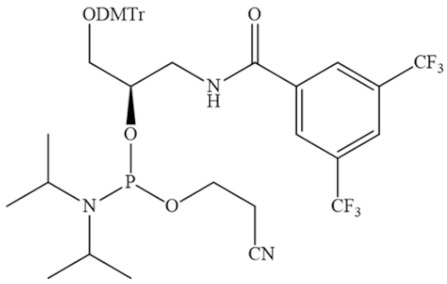	3′- and 5′ ends of aptamer	0.1–1	[147]

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
