# Peer review of "Amphiphilic Oligonucleotide Derivatives—Promising Tools for Therapeutics"

_pharmaceutics, 2024, doi:10.3390/pharmaceutics16111447_

Round 1
Reviewer 1 Report
Comments and Suggestions for Authors
1. Due to the complexity of the review, an abbreviation list will be useful for the readers.
2. Sections 4 and 5 are carefully written, in detail and constitute a complex literature review. On the other hand section 3, should be improved. Maybe one image will increase its quality.
3. A part of the information from sub-section 5.2. Interaction of Lipid-containing Oligonucleotides with Serum Albumin, can be synthesis as a table, in order to be more comprehensive and reader friendly.
Author Response
|
Thank you for taking the time to review this manuscript. You will find detailed responses to your comments below. Appropriate changes can be tracked in the resubmitted manuscript file. They have been made in edit mode. |
|
Comments 1: Due to the complexity of the review, an abbreviation list will be useful for the readers. |
|
Response 1: Thank you for pointing this out. We have added a list of abbreviations introduced in the manuscript. It can now be found on lines 22-28 in the resubmitted manuscript file. |
|
Comments 2: Sections 4 and 5 are carefully written, in detail and constitute a complex literature review. On the other hand, section 3, should be improved. Maybe one image will increase its quality. |
|
Response 2: Thank you for your valuable comment. We have added a corresponding figure in section 3. |
|
Comments 3: A part of the information from sub-section 5.2. Interaction of Lipid-containing Oligonucleotides with Serum Albumin, can be synthesis as a table, in order to be more comprehensive and reader friendly. |
|
Response 3: Thank you for your suggestion. We have added the corresponding table to sub-section 5.2. |

Reviewer 2 Report
Comments and Suggestions for Authors
The manuscript provides an insightful overview of amphiphilic oligonucleotide derivatives, focusing on their therapeutic applications. However, several sections could be strengthened with additional data or specific references to enhance scientific rigor. For Instance,
Line 92: Adding one or two recent studies from the past two years that report on successful clinical applications of RNAi or ASO therapies would strengthen the clinical relevance of these mechanisms. For example, a citation of recent advances in siRNA therapies for liver diseases or other targeted applications would underscore the therapeutic impact.
Line 94: To reflect the effectiveness of these mechanisms in real-world clinical settings, briefly mention a recent breakthrough (e.g., FDA-approved RNAi therapies or promising ASO clinical trials). This will bring the discussion up to date and reinforce the clinical potential of these therapeutic mechanisms.
Line 105: The statement, "nucleases are widely expressed in plasma, particularly 3'-exonucleases," would be more robust with a citation supporting the prevalence and specific impact of 3'-exonucleases on oligonucleotide degradation. Including a recent study that quantifies nuclease activity in plasma would lend more scientific weight here.
Line 113: The sentence stating that “only an indefinitely small quantity is capable of escaping from the vessel lumen and entering the interstitial fluid” suggests a limited distribution of therapeutic oligonucleotides due to vascular barriers. To strengthen this claim, a reference to a study measuring the typical plasma-to-interstitial fluid distribution of oligonucleotides would help substantiate this observation.
Lines 121-122: Lysosomal degradation is a significant challenge post-cell entry, and a reference could be useful. Citing specific studies on lysosomal degradation rates of therapeutic oligonucleotides would provide empirical support and clarify why delivery strategies like lipid conjugation are necessary to overcome this hurdle.
Line 125-127: When stating that solid-phase synthesis allows for the “production of almost any desired ON quickly and economically,” it would be beneficial to briefly mention the scalability of these methods. Including references to studies or reviews that provide production yield rates or cost-benefit analysis data for common modifications could clarify the practical utility of solid-phase synthesis.
Lines 127-129: In this line, a table summarizing the pharmacokinetic outcomes for major types of modifications (e.g., phosphorothioate, 2'-O-methyl) could greatly enhance reader understanding. For example, providing stability data (e.g., half-lives in plasma or tissues) or biodistribution details for each modification type would allow readers to easily compare their effectiveness. Each row could include columns for stability index, typical biodistribution, and known advantages or limitations.
Lines 131-134: When discussing the effects of different modifications (e.g., inter-nucleotide phosphate, ribose backbone), it would strengthen the section to briefly reference specific experimental studies that demonstrate the benefits of each modification. For instance, citing studies that show increased half-life for phosphorothioate modifications or improved cell membrane penetration for lipid modifications would clarify how each chemical modification impacts therapeutic performance.
Lines 238-246: Complex terms like “methylene morpholine rings” or “phosphorodiamidate bonds” might not be familiar to all readers. Consider briefly defining these terms or providing more accessible language to explain their relevance to therapeutic efficacy.
Line 270: Briefly includes one or two recent studies showing how lipid modifications reduce immunogenicity in oligonucleotide therapies. A quick mention of findings on decreased immune response with lipid-ON conjugates compared to unmodified ONs would highlight an essential advantage.
Lines 280-284: Add a short reference to a study that contrasts lipid modifications with other conjugates, highlighting any strengths or limitations (e.g., lipid conjugates may have higher cellular uptake but different tissue specificity compared to peptide conjugates). This quick comparison will give readers a more comprehensive view of why lipids are particularly suited for certain therapeutic applications.
Lines 285-290: Adding a citation or two on how lipid modifications enhance tissue specificity, especially in liver or muscle tissue targeting, would strengthen the discussion on biodistribution. For example, referencing studies on cholesterol or fatty acid conjugates that demonstrate improved targeting could provide a more comprehensive view.
Line 310: Include specific particle size data for typical lipid-ON micelles, such as a range (e.g., 10-30 nm) reported for common lipid types (like cholesterol vs. fatty acid conjugates). This quick addition will clarify the potential impact of micelle size on biodistribution.
Line 316: Briefly mention how different particle sizes influence pharmacokinetics in tissues like the liver or muscle, referencing a study that links particle size to tissue targeting or cellular uptake efficiency. This will provide a clearer picture of why micelle size matters in therapeutic applications.
Line 319: When describing micelle formation, briefly mention a typical particle size range observed for common lipid-ON conjugates (e.g., 10-30 nm) to give readers a clearer picture. Citing a study that includes size data for specific lipid modifications, such as cholesterol vs. fatty acid conjugates, would add context.
Lines 322-325: Include a one-sentence reference to studies that have investigated how micelle size influences cellular uptake efficiency. For instance, a quick mention of size-dependent uptake in liver or muscle tissue would enhance understanding of how these structures interact with different tissues.
Throughout the Manuscript): Terms like “ONs” and “oligonucleotide derivatives” are used interchangeably, which might confuse readers. Consistently using a single term (with the abbreviation “ON” clarified early on) would improve readability.
Figure 5: Including a legend or key for the lipid moieties and oligonucleotide parts in Figure 5 would help readers unfamiliar with chemical structures better understand the schematic.
Author Response
|
Thank you for taking the time to review this manuscript. We have carefully considered the comments and suggestions and revised the manuscript accordingly. We have attached a word file with detailed responses to your comments. The corresponding changes can be tracked in the resubmitted manuscript file. They were made in the editing mode. |
|
Comments 1: Line 92: Adding one or two recent studies from the past two years that report on successful clinical applications of RNAi or ASO therapies would strengthen the clinical relevance of these mechanisms. For example, a citation of recent advances in siRNA therapies for liver diseases or other targeted applications would underscore the therapeutic impact. |
|
Response 1: Thank you for your comment. We have included the necessary references. The relevant changes can be found on lines 90-92 and 104-106 in the resubmitted manuscript file. |
|
Comments 2: Line 94: To reflect the effectiveness of these mechanisms in real-world clinical settings, briefly mention a recent breakthrough (e.g., FDA-approved RNAi therapies or promising ASO clinical trials). This will bring the discussion up to date and reinforce the clinical potential of these therapeutic mechanisms. |
|
Response 2: Thank you for your comment. Similar to the previous comment, we have included the necessary references. |
|
Comments 3: Line 105: The statement, "nucleases are widely expressed in plasma, particularly 3'-exonucleases," would be more robust with a citation supporting the prevalence and specific impact of 3'-exonucleases on oligonucleotide degradation. Including a recent study that quantifies nuclease activity in plasma would lend more scientific weight here. |
|
Response 3: Thank you for your remark. We have included the necessary references. The relevant changes can be found on lines 118-119 in the resubmitted manuscript file. |
|
Comments 4. Line 113: The sentence stating that “only an indefinitely small quantity is capable of escaping from the vessel lumen and entering the interstitial fluid” suggests a limited distribution of therapeutic oligonucleotides due to vascular barriers. To strengthen this claim, a reference to a study measuring the typical plasma-to-interstitial fluid distribution of oligonucleotides would help substantiate this observation. |
|
Response 4: Thank you for your comment. We have included the necessary references. The relevant changes can be found on line 128 in the resubmitted manuscript file. |
|
Comments 5. Lines 121-122: Lysosomal degradation is a significant challenge post-cell entry, and a reference could be useful. Citing specific studies on lysosomal degradation rates of therapeutic oligonucleotides would provide empirical support and clarify why delivery strategies like lipid conjugation are necessary to overcome this hurdle. |
|
Response 5. Thank you for your observation. We have added additional information and relevant references. The relevant changes can be found on lines 133-138 in the resubmitted manuscript file. |
|
Comments 6. Line 125-127: When stating that solid-phase synthesis allows for the “production of almost any desired ON quickly and economically,” it would be beneficial to briefly mention the scalability of these methods. Including references to studies or reviews that provide production yield rates or cost-benefit analysis data for common modifications could clarify the practical utility of solid-phase synthesis. |
|
Response 6. Thank you for your comment. We have included the necessary references. |
|
Comments 7. Lines 127-129: In this line, a table summarizing the pharmacokinetic outcomes for major types of modifications (e.g., phosphorothioate, 2'-O-methyl) could greatly enhance reader understanding. For example, providing stability data (e.g., half-lives in plasma or tissues) or biodistribution details for each modification type would allow readers to easily compare their effectiveness. Each row could include columns for stability index, typical biodistribution, and known advantages or limitations. |
|
Response 7. Thank you for your suggestion. We have added additional information and the corresponding table to section 4. The relevant changes can be found on lines 164-173 in the resubmitted manuscript file. |
|
Comments 8. Lines 131-134: When discussing the effects of different modifications (e.g., inter-nucleotide phosphate, ribose backbone), it would strengthen the section to briefly reference specific experimental studies that demonstrate the benefits of each modification. For instance, citing studies that show increased half-life for phosphorothioate modifications or improved cell membrane penetration for lipid modifications would clarify how each chemical modification impacts therapeutic performance. |
|
Response 8. Thank you for your comment. We have added summary information and relevant references. The relevant changes can be found on lines 155-161 in the resubmitted manuscript file. |
|
Comments 9. Lines 238-246: Complex terms like “methylene morpholine rings” or “phosphorodiamidate bonds” might not be familiar to all readers. Consider briefly defining these terms or providing more accessible language to explain their relevance to therapeutic efficacy. |
|
Response 9. Thank you for pointing this out. We have added a figure explaining the terms used. The relevant changes can be found on lines 276-277, 290-291 in the resubmitted manuscript file. |
|
Comments 10. Line 270: Briefly includes one or two recent studies showing how lipid modifications reduce immunogenicity in oligonucleotide therapies. A quick mention of findings on decreased immune response with lipid-ON conjugates compared to unmodified ONs would highlight an essential advantage. |
|
Response 10. Thank you for your remark. We have included additional information and the necessary references. The relevant changes can be found on lines 313-317 in the resubmitted manuscript file. |
|
Comments 11. Lines 280-284: Add a short reference to a study that contrasts lipid modifications with other conjugates, highlighting any strengths or limitations (e.g., lipid conjugates may have higher cellular uptake but different tissue specificity compared to peptide conjugates). This quick comparison will give readers a more comprehensive view of why lipids are particularly suited for certain therapeutic applications. |
|
Response 11. Thank you for your remark. We have included additional information and the necessary references. The relevant changes can be found on lines 330-335 in the resubmitted manuscript file. |
|
Comments 12. Lines 285-290: Adding a citation or two on how lipid modifications enhance tissue specificity, especially in liver or muscle tissue targeting, would strengthen the discussion on biodistribution. For example, referencing studies on cholesterol or fatty acid conjugates that demonstrate improved targeting could provide a more comprehensive view. |
|
Response 12. Thank you for your comment. This information can be found on lines 347-349 and 351-353 in the resubmitted manuscript file. |
|
Comments 13. Line 310: Include specific particle size data for typical lipid-ON micelles, such as a range (e.g., 10-30 nm) reported for common lipid types (like cholesterol vs. fatty acid conjugates). This quick addition will clarify the potential impact of micelle size on biodistribution. |
|
Response 13. Thank you for your suggestion. We have included additional information and the necessary references. The relevant changes can be found on lines 367-369 in the resubmitted manuscript file. |
|
Comments 14. Line 316: Briefly mention how different particle sizes influence pharmacokinetics in tissues like the liver or muscle, referencing a study that references particle size to tissue targeting or cellular uptake efficiency. This will provide a clearer picture of why micelle size matters in therapeutic applications. |
|
Response 14. Thank you for your remark. We have included additional information. The relevant changes can be found on lines 380-381 and 386-390 in the resubmitted manuscript file. |
|
Comments 15. Line 319: When describing micelle formation, briefly mention a typical particle size range observed for common lipid-ON conjugates (e.g., 10-30 nm) to give readers a clearer picture. Citing a study that includes size data for specific lipid modifications, such as cholesterol vs. fatty acid conjugates, would add context. |
|
Response 15. Thank you for your comment. Similar to the comment 13, we have included the necessary references. |
|
Comments 16. Lines 322-325: Include a one-sentence reference to studies that have investigated how micelle size influences cellular uptake efficiency. For instance, a quick mention of size-dependent uptake in liver or muscle tissue would enhance understanding of how these structures interact with different tissues. |
|
Response 16. Thank you for your comment. Similar to the comment 14, we have included the necessary references. |
|
Comments 17. Throughout the Manuscript): Terms like “ONs” and “oligonucleotide derivatives” are used interchangeably, which might confuse readers. Consistently using a single term (with the abbreviation “ON” clarified early on) would improve readability. |
|
Response 17. Thank you for pointing this out. The terms 'oligonucleotide derivatives' or 'ON derivatives' refer specifically to oligonucleotides modified at various positions. The term 'ON' is simply an abbreviation of the word oligonucleotide and when used alone refers to unmodified oligonucleotides. |
|
Comments 18. Figure 5: Including a legend or key for the lipid moieties and oligonucleotide parts in Figure 5 would help readers unfamiliar with chemical structures better understand the schematic. |
|
Response 18. Thank you for your suggestion. We have added the necessary explanations to the figure. |

Reviewer 3 Report
Comments and Suggestions for Authors
Bauer and Dmitrienko review on the use of chemically modified nucleic acid derivatives for therapeutic applications. The particular focus is on oligonucleotides conjugated to lipid moieties. The review article considers the latter as one of the strategies used to date to improve the pharmacodynamic and pharmacokinetic characteristics of therapeutic oligonucleotides. The manuscript is carefully designed and covers different chemical modifications of oligonucleotides (e.g., phosphate and sugar backbone modifications, etc.) as well as modifications via attachment of hydrophobic residues to the oligonucleotide chains. The review is well-written, comprehensive, and systematic and contains description of various systems as well as challenges and perspective. I have very few and mostly technical recommendations and suggestions for improvements that are listed below.
- Avoid multiple introductions of abbreviations. E.g., ASO is introduced several times.
- In some of the figures (Figure 9, Figure 15) I don’t see the letters for the substituents given in the figure caption.
- In Figure 14 check and correct the formula of the conjugate. In particular, the oxyethylene units (those in brackets with degrees of polymerization of 4 and 45) are wrongly presented.
Author Response
|
Thank you for taking the time to review this manuscript. You will find detailed responses to your comments below. Appropriate changes can be tracked in the resubmitted manuscript file. They have been made in edit mode. |
|
Comments 1: Avoid multiple introductions of abbreviations. E.g., ASO is introduced several times. |
|
Response 1: Thank you for pointing this out. We have removed multiple introductions of abbreviations. |
|
Comments 2: In some of the figures (Figure 9, Figure 15) I don’t see the letters for the substituents given in the figure caption. |
|
Response 2: Thank you for your remark. We have made the necessary corrections in the figure captions. |
|
Comments 3: In Figure 14 check and correct the formula of the conjugate. In particular, the oxyethylene units (those in brackets with degrees of polymerization of 4 and 45) are wrongly presented. |
|
Response 3: Thank you for your comment. We have made the necessary corrections to the chemical formula of the conjugate in Figure 14. |
